# First Report on the Presence of Toxic Metals and Metalloids in East Asian Bullfrog (*Hoplobatrachus rugulosus*) Legs

**DOI:** 10.3390/foods11193009

**Published:** 2022-09-27

**Authors:** Emanuela Bacchi, Gaetano Cammilleri, Marina Tortorici, Francesco Giuseppe Galluzzo, Licia Pantano, Vittorio Calabrese, Mariagrazia Brunone, Antonio Vella, Andrea Macaluso, Gianluigi Maria Lo Dico, Vincenzo Ferrantelli

**Affiliations:** 1Istituto Zooprofilattico Sperimentale della Sicilia “A. Mirri”, Via Gino Marinuzzi 3, 90129 Palermo, Italy; bacchiemanuela@gmail.com (E.B.); tortorici.marina@gmail.com (M.T.); francescogiuseppe92@gmail.com (F.G.G.); licia.pantano@izssicilia.it (L.P.); laboratorio.residui@gmail.com (A.V.); andrea.macaluso@izssicilia.it (A.M.); gigilodico@gmail.com (G.M.L.D.); vincenzo.ferrantelli@izssicilia.it (V.F.); 2Dipartimento di Scienze Biomediche e Biotecnologiche, Università Degli Studi di Catania, Torre 11 Biologica Via Santa Sofia 97, 95123 Catania, Italy; calabres@unict.it; 3Freelance Biologist, 90142 Palermo, Italy; maria.grazia83@hotmail.it

**Keywords:** toxic metals, arsenic, edible frogs, ICP-MS

## Abstract

We examined the presence of As, Cr, Cd, and Pb in 42 samples of farmed East Asian bullfrog (*Hoplobatrachus rugulosus*) from Vietnam and Thailand by inductively coupled plasma-mass spectrometry (ICP-MS). An estimation of the dietary intake and exposure to the toxic elements analysed was also carried out. The results showed very high As levels, with mean values of 0.094 ± 0.085 mg/Kg w.w. and a maximum of 0.22 mg/Kg. No significant differences were found for As contents between areas of production (*p* > 0.05). No detectable Cd contents were found in all the samples examined. The Pb concentrations of the East Asian bullfrog legs samples were below the European Commission’s permitted levels. The Cr and Pb contents of the East Asian bullfrog produced in Vietnam were significantly higher than that produced in Thailand (*p* < 0.05). The target hazard quotient (THQ) ratio for Cr was not exceeded for all the samples analysed. In contrast, the benchmark dose lower confidence limit (BMDL) and THQ ratios for As were exceeded, indicating carcinogenic and non-carcinogenic risks for those who consume this type of food. The results of this work confirm the role of As-contaminated water absorption as an important source of arsenic for these adult organisms.

## 1. Introduction

The edible frog is a food product most widely consumed in France and Italy, among European countries, although consumption in Italy is lower than the European average (8000–9000 tons/year, one of the highest in the world). The portion consumed, and marketed, is usually represented by the hind legs as they have a higher percentage of muscle mass than the trunk and front legs.

This growing demand has led to the implementation of effective frog farming systems. Frog farming is becoming profitable as it is possible to breed frogs in a limited space with a small amount of water and their byproducts have a higher market value than most freshwater fish [1].

The most widely consumed European frog is the green frog (Pelophylax spp.), mostly in France and Belgium leading to severe local extinctions [2].

To date, the species marketed by non-European Countries are difficult to identify because the product is commercialized as skinned or processed, and labeling is often incorrect [1].

Recent studies on *Rana esculenta* species have shown heavy metal values (aluminum, cadmium, copper, lead, chromium, nickel, boron, silicon, and zinc), which are below the tolerable daily intake (TDI) [2].

In another study, the number of heavy metals in the marsh frog *Pelophylax ridibundus* species was analysed, showing Pb and Cd contents below the limits allowed by European Union legislation [3]. However, few works on toxic metals accumulation in bullfrogs are still present.

The East Asian bullfrog (*Hoplobatrachus rugulosus*) is a species of frog of the Dicroglossidae family. It can live in terrestrial or semi-terrestrial habitats but often can be found at the edges of rice fields [4]. At present, there is no data in the literature regarding the breeding cycle of frogs of the species *Hoplobatrachus rugulosus*, growth times, feeding protocols, and health control. Furthermore, there is no evidence of the presence of toxic substances, such as toxic metals and metalloids, considering that this species is of considerable economic and commercial interest.

At present, trace elements can be determined in food matrices with good precision and sensitivity by ICP-MS method [5,6,7].

The aim of this work is to assess the presence of As, Cr, Cd, and Pb in East Asian bullfrog (*Hoplobatrachus rugulosus*) legs marketed in the large-scale retail trade. The study was performed for three reasons: (i) to provide the first data on the toxicological aspects of the edible frogs, (ii) to make an initial risk assessment on the consumption of the frogs, and to provide scientific evidence needed to raise awareness in the legislature.

## 2. Materials and Methods

### 2.1. Sample Collection and ICP-MS Analysis

Forty-two frozen frog leg samples of the East Asian bullfrog *(Hoplobatrachus rugulosus*) were purchased from international e-commerce sites in 2021. All the samples came from farms in Vietnam (n = 21) and Thailand (n = 21) and were produced in 2019, as stated on the label. The procurement of the samples was done in order to obtain a balanced number by country of production. The samples were digested according to protocols reported previously [3,4].

Briefly, 1 ± 0.01 g of the frog legs samples were put into poly-tetrafluoroethylene-tetrafluoroethylene vessels with 3 mL of 60% (*V*/*V*) ultrapure nitric acid and 5 mL of deionised water and digested using an UltraWAVE digestion system (Milestone, Sorisole, Italy).

The digested samples were made up to a volume of 50 mL with ultrapure water, and filtered with 0.45 µm filters before ICP-MS analysis. The As, Cr, Cd, and Pb determination were carried out using a 7700× series ICP-MS (Agilent Technologies, Santa Monica CA, USA). The operating conditions and instrumental settings were as follows: RF-Power 1550 W, reflect power < 30, carrier gas flow 1.0 mL min^−1^, plasma gas flow 15 L min^−1^, auxiliary gas flow 0.9 mL min^−1^, spray chamber temperature +2 °C, lens voltage 6.25 V, mass range 6–220 a.m.u., mass resolution 0.7, integration time 3 point ms^−1^, 3 points of peak, and 4 replicates. The analysis was performed using calibration curves, constructed by linear interpolation of at least 7 points corresponding to the readings of the standard solutions and the calibration blank, allowing a maximum error of 5% on the reading of individual standards and a correlation coefficient r^2^ > 0.999.

The ICP-MS method was validated according to the European Decision 657/2002 and the ISO 17025:2018 for calibration, precision, and trueness by recovery. The trueness of the method was evaluated by calculating the recovery at 3 concentration points using the formula:recovery% = 100C/(diluted concentration)(1)
where C is the concentration obtained of the analyte, allowing a range between 90 and 110%. Intra-laboratory repeatability and reproducibility were calculated as percent coefficient of variation (CV%) by analysing the same samples used for the recovery calculation on three different days. The limits of detection and quantification (LOD and LOQ), were determined by the 3σ and 10σ approach according to Lo Dico et al. (2018) [5].

### 2.2. Data Collection and Statistical Analysis

The As, Cd, Cr, and Pb results were expressed as mg/kg wet weight (w.w.). All the results below the LOQ of the method were considered as half of the LOQ for the statistical analysis, taking into account Helsel (2005) [6]. The normality of the data groups of the samples was verified with the Shapiro–Wilk test prior to analysis. The conditions of normal distribution were not met, therefore, a Wilcoxon rank sum test was carried out to evaluate significant differences between production areas (Vietnam vs. Thailand).

### 2.3. Health Risk Assessment

The estimated daily intake (EDI) and the estimated weekly intake (EWI) were calculated to estimate the exposure to As, Cr, Cd, and Pb in a short time. The assumption was made that 225 g of frog legs are consumed daily. The following formula was used for the calculation:
*EDI = (DFC × C)/BW*
(2)

where *C* is the concentration of the analyte (As, Cr, Cd, and Pb) in the product analysed (mg/kg), *DFC* is the daily food consumption rate (225 g/day), and *BW* is the average body weight (adult, 70 kg). Long-term exposure to As and Pb was estimated using the benchmark *BMDL* index using the equation [7]:
*BMDL = EDI/BW*
(3)


BMDL values for As are 3 μg/kg/BW/day, with a benchmark dose lower confidence of 0.5% (BMDL0.5) according to FAO/WHO standards [8].

Furthermore, the Target Hazard Quotient (THQ index) for As, Cr, and Cd was calculated according to the following equation:
*THQ = (EF × ED × P × C)/(RfD × BW × T) ×* 10^−3^(4)
where EF is the frequency of exposure (365 days/year), *ED* is the duration of exposure (70 years), equivalent of the average lifetime, *P* is the portion of the analysed product consumed daily according to Mani et al. 2021 (225 g) [9], *C* is the concentration of As, Cr, and Cd in the analysed product (mg/kg), *RfD* is the oral reference dose (1 × 10^−3^, 3 × 10^−3^ and 3 × 10^−4^ mg/kg body weight/day for Cd, Cr and As, respectively, according to the guidelines of the United States Environmental Protection Agency (US EPA)), BW is the body weight, as mentioned above, *T* is the average time of exposure for non-carcinogens (365 days/year × 30 years = 10,950 days). *THQ* has been applied to analyse the potential non-cancerogenic effect of the metals present in food. If the *THQ* > 1, it may indicate a potential risk related to the consumption of toxic metal and metalloids with the product analysed. A *THQ* < 1 is associated with a low non-cancerogenic risk [10,11]. The health risk assessment for As was carried out according to the scientific opinion of the European Food Safety Authority, [12] which assumed that 70 percent of the total arsenic found in non-marine foods existed as inorganic arsenic (iAs).

## 3. Results and Discussion

The ICP-MS calibration test for As, Cr, Cd, and Pb gave satisfactory results (r^2^ > 0.999; Table 1). The LOD and LOQ values obtained were in accordance with the regulation (CE) n. 333/2007 and Commission Decision 2002/657/EC (Table 1). Recoveries evaluation showed values between 96% and 105%.

Cr, As, and Pb contents were found in 45.2%, 61.9%, and 64.3% of the samples examined, respectively. The mean contents of trace elements followed the order As > Cr > Pb > Cd. As was the most abundant element, showing a mean of 0.094 ± 0,084 mg/kg and a maximum of 0.222 mg/kg w.w., followed by Cr with 0.0176 ± 0,022 mg/kg w.w. and a maximum of 0.116 mg/kg w.w. Conversely, Pb was the less abundant element, with mean contents of 0.0170 mg/kg ± 0,127 mg/kg and a maximum of 0.0382 mg/kg. No detectable Cd levels were found in all the samples examined. The maximum Cr, As, and Pb values found in the samples came from Thailand (0.116 e 0.222 and 0.0382 mg/kg, respectively). The samples from Vietnam showed significantly higher Cr and Pb contents than the samples from Thailand, with mean values of 0.025 ± 0.028 mg/Kg (W = 0.757; *p*-value = 0.00015) and 0.022 ± 0.011 mg/kg (W = 131, *p*-value = 0.02194), respectively. No significant differences were found for As contents (W = 171, *p*-value = 0.2021). Regarding the dietary exposure assessment, the EDI, EWI, BMDL, and THQ for the samples analysed are shown in Table 3.

The BMDL for iAs ranged from 9.6 × 10^−5^ to 0.0071 μg/kg/BW/day, exceeding the toxicological guidance value set by the Codex Alimentarius Commission (133% of the norm). Even the THQ determination showed average values above the threshold value. The THQ index for Cr amounted from 0.011 to 0.291, remaining lower than the threshold value.

The results of this work showed hazardous iAs contents in the edible parts (legs) of the East Asian bullfrog, suggesting potential risks for the consumers’ health.

Frogs are suitable bioindicators of environmental pollution, detecting harmful levels of trace elements in the environment [13,14]. Like other amphibians, frogs live part of their cycle in aquatic environments, taking up trace elements from river water, and accumulate them in the body tissues, not having developed effective detoxification mechanisms [15]. Different works confirmed that the bioaccumulation factor for heavy metals in frogs as a result of uptake from the water was greater than that from the soil. Thus, frogs have a much greater metal bioaccumulation rate from the surrounding water than from the soil/sediment [16,17]. Metal accumulation by frogs’ tissues has been previously reported in several studies. Thus, Loumbourdis and Wray (1998) [18] reported high levels of Cr and Cd in the liver of *P. ridibundus* inhabiting a small river in Macedonia, Northern Greece. Our findings showed significant differences between the production areas (Vietnam and Thialand) for Cr and Pb. The East Asian bullfrog samples from Vietnam showed higher mean values of Cr than the East Asian bullfrog from Thailand but still lower than those found by Mani et al. 2021 [9] in the Marsh frog (*Pelophylax ridibundus*) from Kalasin (Thailand; 2.782 mg/kg), this difference could be due to the environmental pollution status of the two areas, considering the bioaccumulation capacity of the frogs. Our As and Pb results are comparable to those reported by Zehelev et al. (2020) [2] in the adult forms of *Pelophylax ridibundus* belonging to unpolluted reference sites in southern Bulgaria; on the contrary, the average values found in *Pelophylax ridibundus* of recognised polluted sites presented levels 4.5 times higher for As and 25 times higher for Pb than our study. No detectable Cd contents were found in the East Asian bullfrog samples examined, in contrast to what was found in other frog species from Eastern Europe and East Asia [2,18,19]. The comparisons of our results with other studies reported in literature [2,19,20] seem to contradict the negative relationship between the length-weight of frogs and Cd concentration verified by Mani et al. (2021) [9]. However, Mani et al. (2021) [9] stated that this correlation was not statistically significant.

International authorities such as the European Commission have set the maximum allowable concentration limits for Cd and Pb. None of the East Asian bullfrog samples analysed exceeded the Pb and Cd threshold imposed by the EC regulation 1881/2006.

This study also provides an estimate of the dietary intake and examines the dietary exposure to Cr and iAs through the consumption of farmed East Asian bullfrog. However, it should be noted that the risk assessment for iAs was carried out considering the assumption scenarios relating to the proportion of total arsenic representing inorganic arsenic, based upon reasonable assumptions concerning arsenic speciation of various foods obtained from EFSA monitoring and in the scientific literature. This encourages further studies based on the use of analytical techniques capable of performing arsenic speciation such as HPLC coupled with ICP-MS [21]. The Cr concentration in the edible parts of the frogs examined did not exceed the guidelines and standards of World Health Organization/Food and Agricultural Organization: WHO/FAO [7]. Since food is a major source of exposure to Cr [22], monitoring the quantities entering the trophic dynamics of the food chain can provide useful data on the accumulation rates of this metal in edible frogs.

Feeding is the critical process in culturing frogs successfully. Frogs reared outdoors will obtain some natural foods, but for the intensive commercial culture of frogs in high densities, supplemental food must be supplied [23]. Considering this, the low Cr and Cd contents found could be due to the use of controlled feeding strategies by farmers.

In contrast, the results of BMDL calculation for As have shown that the threshold values have been exceeded by 54.8% of the samples examined. Furthermore, the mean THQ value obtained suggests adverse health effects on adult consumers, showing values above 1 for 57% of the samples examined. The inorganic form of As is listed by the IARC (Group 1) as a carcinogen [24]. Exposure to As has a toxic effect on diseases of the nervous system, children’s nervous development, respiratory system, and skin.

These alarming findings could be due to the pollution condition of the farming environment. Previous studies revealed that an elevated accumulation of As in amphibious may be due to the high As content in soil and excessive As-contaminated water. Moriarty et al. (2013) [25] stated that during development, frogs may initially accumulate As at higher concentrations than the As concentrations in their surrounding water, suggesting that As absorption via contaminated water is the likeliest main source of inorganic As to frogs.

## 4. Conclusions

As far as we know, the present work reports the first data on the presence of toxic metals and metalloids in East Asian bullfrog legs. The Cr and Pb concentrations in the edible tissues of East Asian bullfrog showed significant differences according to the area of production as a result of the different pollution status of the two areas considered. The Cd and Pb contents in the edible tissues were below the allowable limits set by the European legislation. However, significantly elevated As values were found, probably related to the state of contamination of the soils and water of the farming environment. The high THQ values calculated for As confirmed that the East Asian bullfrog intake of the production areas considered could cause possible adverse health effects in humans. However, given the small number of samples analysed and the lack of detailed information regarding the breeding methods, further investigations based on type of farming are essential in order to elucidate metal accumulation issues in frequently consumed frogs.

## Figures and Tables

**Table 1 foods-11-03009-t001:** ICP-MS calibration results of the ICP-MS method and LOD-LOQ values of the trace elements analysed.

Element	r^2^	y = mx + q	LOD (µg/kg)	LOQ (µg/kg)
As	1.0000	y = 0.0228 × x + 0.0336^−5^	5.74	6.64
Cd	1.0000	y = 0.0047 × x + 1.9682^−6^	5.53	5.74
Pb	1.0000	y = 0.0433 × x + 8.0581^−5^	5.48	5.76
Cr	0.9996	y = 0.0865 × x + 6.4852^−4^	8.04	9.12

The results of the samples examined are shown in Table 2.

**Table 2 foods-11-03009-t002:** The level of As, Cr, Cd, and Pb (mg/kg) of samples detected in the study East Asian bullfrog. Cd was not detected in any sample. Entries n.d. = not detected.

	As			Cr			Cd			Pb		
	Mean ± SD	Min	Max	Mean ± SD	Min	Max	Mean ± SD	Min	Max	Mean ± SD	Min	Max
Thailand	0.072 ± 0.088	n.d.	0.222	0.009 ± 0.01	n.d.	0.045	n.d.	n.d.	n.d.	0.012 ± 0.013	n.d.	0.038
Vietnam	0.116 ± 0.077	n.d.	0.213	0.025 ± 0.028	n.d.	0.116	n.d.	n.d.	n.d.	0.022 ± 0.011	n.d.	0.038
Total	0.094 ± 0.084	n.d.	0.222	0.018 ± 0.022	n.d.	0.116	n.d.	n.d.	n.d.	0.017 ± 0.013	n.d.	0.038

**Table 3 foods-11-03009-t003:** Average EDI, BDML and THQ values of the samples analysed sorted by area of production. Site 1 = samples from Thailand; Site 2 = samples from Vietnam. NA = not applicable; iA = inorganic arsenic.

	EDI			BMDL			THQ		
	Cr	iAs	Pb	Cr	iAs	Pb	Cr	iAs	Pb
Site 1	0.035 ± 0.037	0.163 ± 0.199	0.040 ± 0.041	NA	0.002 ± 0.003	NA	0.025 ± 0.026	1.269 ± 1.549	NA
Site 2	0.09 ± 0.099	0.26 ± 0.173	0.069 ± 0.036	NA	0.004 ± 0.002	NA	0.064 ± 0.069	2.023 ± 1.343	NA
Total	0.057 ± 0.071	0.211 ± 0.190	0.054 ± 0.041	NA	0.0031 ± 0.0027	NA	0.044 ± 0.056	1.646 ± 1.481	NA

## Data Availability

Data is contained within the article.

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
