# Peer review of "First Report on the Presence of Toxic Metals and Metalloids in East Asian Bullfrog (Hoplobatrachus rugulosus) Legs"

_foods, 2022, doi:10.3390/foods11193009_

Round 1

Reviewer 1 Report

1. Please give more information on the sampling time and the age of samples. These factors would inflence the results. In a word, how to exclude the variables. The number of samples is too small, how to get the representative conclusion?

2. Some literature should be added in the introduction section. In addition, please expand the introduction.

3. Their results showed very high As levels, as we all know, the toxicity of As mainly dependent on their speciation, I strongly recommend that the authors should text the As speciation to better assess their health risks.

4. The health risk assessment based on the total concentrations of heavy metals was widely demonstrated to overestimate the risk. The bioaccessibility or bioavailability should be done to accurately evaluate the risks.

5. Please give more scientific explanation for the results.

6. Authors confirmed that the role of As-contaminated water absorption as an important source of arsenic for these adult organisms, however, I can not find the direct evidence.

7. “The samples from Vietnam showed significantly higher Cr and Pb contents than the samples from Thailand”, please tell the possible reason to the readers. Please give more discussions.

8. In lines 222-224, “The high THQ values calculated for As confirmed that the East Asian bullfrog intake of the production areas considered could cause possible adverse health effects in humans. “, how about the synergistic effects of several heavy metals in the legs?

9. The English-writting should be polished by a native speaker.   

Author Response

  1. Please give more information on the sampling time and the age of samples. These factors would inflence the results. In a word, how to exclude the variables. The number of samples is too small, how to get the representative conclusion?

Dear reviewer, we tried to add more information about the sampling plan carried out; unfortunately, given the fact that this species of frog is not produced in Europe but it is normally marketed through e-commerce systems, we were able to comply only with the information reported on the label. We are strongly in accordance with you regarding the low number of samples analysed, however, as there is still no scientific evidence on the presence of toxic metals and metalloids in this species, we believe that this short communication can provide the first points for future investigations.

  1. Some literature should be added in the introduction section. In addition, please expand the introduction.

Dear reviewer, we tried to expand the introduction within the limits imposed for the short communications format.

  1. Their results showed very high As levels, as we all know, the toxicity of As mainly dependent on their speciation, I strongly recommend that the authors should text the As speciation to better assess their health risks.

Dear reviewer, we really appreciate your precious comment for a comphrensive understanding of the results obtained. On this basis, we decided to carry out a risk assessment according to the scientific opinion of the European Food Safety Authority. More details are discussed in the materials and methods section.

  1. The health risk assessment based on the total concentrations of heavy metals was widely demonstrated to overestimate the risk. The bioaccessibility or bioavailability should be done to accurately evaluate the risks.

Dear reviewer, we reworked the risk assessment based on the previous comment. As this is a short communication, we reserve the right to consider this aspect for future works.

  1. Please give more scientific explanation for the results.

Dear reviewer, we added more explanation in the main document.

  1. Authors confirmed that the role of As-contaminated water absorption as an important source of arsenic for these adult organisms, however, I can not find the direct evidence.

Dear reviewer, we added more information on this aspect in the main document.

  1. “The samples from Vietnam showed significantly higher Cr and Pb contents than the samples from Thailand”, please tell the possible reason to the readers. Please give more discussions.

Dear reviewer, since the sampling was carried out directly in e-commerce sites, as stated in the materials and methods section, so we were not able to have more details about the production areas and therefore give more explanation about the differences between Vietnam and Thailand frogs.

  1. In lines 222-224, “The high THQ values calculated for As confirmed that the East Asian bullfrog intake of the production areas considered could cause possible adverse health effects in humans. “, how about the synergistic effects of several heavy metals in the legs?

Dear reviewer, could you please give more details about this topic? Maybe do you mean if the combination between the toxic metals under study could cause adverse effects?

  1. The English-writting should be polished by a native speaker.

Dear reviewer, we have revised the English form using a professional English editing software, according to your useful suggestion.

Reviewer 2 Report

I have read a manuscript entitled “First report on the presence of toxic metals and metalloids in East Asian bullfrog (Hoplobatrachus rugulosus) legs” The contribution to food safety is worthy. That said, the presentation of the work need to do. The following are this reviewer's assessments

INTRODUCTION

Needs overhaul with more background materials.

Lines 28-36: The first three paragraphs should be re-written and combined into one.

Line 37: “heavily exploited” should be replaced with “mostly”

Line 39: It is not clear what “European foreign-derived trade” means.

Line 42: Is Rana esculenta a different species of bullfrog? This is not clear.

Line 42: The word “interesting” is casual, adds no values and should be removed.

Line 45: Is Pelophylax ridibundus a different species of bullfrog? This is not clear.

Lines 49-50: This sentence needs a rewrite “It can live in terrestrial or semi terrestrial habitats or at the edges of rice fields.”  “At the or at the edges of rice fields” is out of place.

Line 56: “…, considered the election method for the detection 56 of toxic elements” is confusing. Authors should consider deleting.

Lines 58-62: For readability, breakdown this long sentence. Perhaps the following rewrite can be of help “The aim of this work is to assess the presence of As, Cr, Cd and Pb in East Asian bullfrog (Hoplobatrachus rugulosus) legs marketed in the large-scale retail trade. This is done for three reasons: (i) to provide first data on the toxicological aspects of the edible frogs, (ii) to make an initial risk assessment on the consumption of the frogs, and to provide scientific evidence needed to raise awareness in the legislature.”

Line 69: Mineralisation procedure could mean just about anything. Seriously consider replacing the entire sentence with something like “The samples were digested according to protocols reported before [8,9].”

Line 74: Instead of “The samples mineralised were …” consider rewriting as “The digested samples were….”

Line 75: Consider changing “before the ICP-MS” to “before ICP-MS”

Line 77: Consider changing “…instrumental settings were set as following” to “…instrumental settings were as follows”

Lines 81-84: Confusing and must be rewritten. Furthermore, (i) at least in the “at least 7 points” points to uncertainty; (ii) “admitting” in line 83 (and line89) are really out of place; and (iii) what is “white calibration,” how does it apply to this study?

Line 85: Which method was validated, sample collection or ICP-MS or both?

MATERIALS AND METHODS

A.     The foundation of Health Risk Assessment (HRA), as reflected in this manuscript, is confusing and shows limited understanding of risk assessment of mixed chemicals as exposed by the US EPA. There are four (4) sequential steps required to carry out a successful HAR; this knowledge is not clear in this manuscript. This reviewer recognizes this shortcoming as probably a consequence of difficulties associated with translating text from other languages to written English (see Acknowledgments line 236). But, given that the gist of this paper is about Health Risk Assessment and the THQ system, it is strongly recommended that the authors include reference to the very simple description of these two concepts, such as those provided in https://doi.org/10.1371/journal.pone.0212938 and https://doi.org/10.1016/j.toxrep.2017.03.006 for the benefit of readers encountering the concept of HRA and THQ for the first time.”

B.      Consistency is required. For example, the authors should decide whether to use:

(i)     “where” (equation 2), “Where” (line 89 and equation 3) or “Where:” (equation 4)

(ii)   “As, Cr and Cd” (Line 124) or “cadmium, chromium and arsenic” (Line 126).

(iii) “kg” (line 116), or “Kg” (line 96 and the entire manuscript). NB: The symbol of the SI unit of mass kilogram is “kg”, not Kg. This correction, in this reviewer’s opinion, must be made throughout the manuscript.

C.      Remove unnecessary repetition. For example,

(i)     “BW is the average body weight (70 Kg)” is found on lines 110, 115 and 127 on the same page. Please define whose body weight is being referred to here: male, female, children, adult human, European?

(ii)   “EDI is the estimated daily intake” is found on lines 104 and 115 on the same page.

D.     From equation (2), EDI is proportional to 1/BW and from equation (3), BMDL is therefore proportional to 1/BW2. Is this the message the authors are conveying?

E.      The “estimated weekly intake (EWI)” in Line 104 is not a category in Table 3 as it is incorrectly stated in Line 155.

Line 98: (Helsel (2005) [11]” is incomplete in the References section.

Line 99: “…prior to analysis.” Are the authors refereeing to data or samples analysis?

Lien 106:  “… 225g of frog legs are consumed daily” Who is consuming this? It is not clear from text.

RESULTS AND DISCUSSION

A.     Consistency is required. For example, the authors should decide whether to use:

(i)     Small “t” or capital “T” when referring to a table in text (e.g., “table 1” & “table 2” (lines 135, 136 & 140), but Table 3” line 155. 

B.      Consider rewriting “The linearity test of the ICP-MS method for As,Cr, Cd and Pb gave satisfactory results” as “The linearity test of the ICP-MS method for As, Cr, Cd and Pb gave satisfactory results”

C.      Table 2: (i) Replace “Site 1” and “Site 2” in table and caption with “Thailand” and “Vietnam” respectively. No need to be complicated since the expressions “Site 1” and “Site 2” are not used by the authors in the text.  (ii) For clarity, consider rewriting the caption to read “The level of As, Cr, Cd and Pb (mg/kg) of samples detected in the study East Asian bullfrog. Cd was not detected in any sample. Entries n.d.= not detected.” If authors agree to adding the statement “Cd was not detected in any sample,” the entire and redundant Cd columns in Table 2 can be deleted.

D.     Lines 134-164: It is difficult to follow. Advice: Sort similar themes into separate paragraphs or subheadings.

Lines 134-139: Consider rewriting this paragraph and moving it to the “Materials and Methods– along with Table 1 under a separate subsection; you may call the subsection “2.3 Calibration, limits of detection and quantification.   The authors must state briefly how LOD & LOQ are determined here.  Of note, “Linearity test” is vague; consider replacing it with “ICP-MS calibration” both in text and in Table 1 caption. Also in the caption to Table 1, define “x” and “y”. In Line 136, “Recoveries evaluation showed values between 96% and 105%”, provide what is being evaluated and reason for evaluation.

Line 147: What is “Conservely”?

Line 149: New paragraph should begin at “The maximum Cr, As e Pb values found in the…” What is “e” between AS and Pb?

CONCLUSION

Acceptable.

REFERENCES

Lines 276-277: Incomplete. Must be cited in full as required by the journal.

Author Response

Lines 28-36: The first three paragraphs should be re-written and combined into one.

Dear reviewer, we modified the paragraphs according to your precious suggestion.

Line 37: “heavily exploited” should be replaced with “mostly”

Dear reviewer, we modified this word according to your suggestion.

Line 39: It is not clear what “European foreign-derived trade” means.

Dear reviewer, we modified this part with “non-European countries”

Line 42: Is Rana esculenta a different species of bullfrog? This is not clear.

Dear reviewer, we confirm that Rana esculenta is not a bullfrog. Most of the studies on metal accumulation reported in literature are focused on this frog species.

Line 42: The word “interesting” is casual, adds no values and should be removed.

Dear reviewer, we removed this word according to your precious suggestion

Line 45: Is Pelophylax ridibundus a different species of bullfrog? This is not clear.

Dear reviewer, we specified it in the main document.

Lines 49-50: This sentence needs a rewrite “It can live in terrestrial or semi terrestrial habitats or at the edges of rice fields.”  “At the or at the edges of rice fields” is out of place.

Dear reviewer we rewrote this sentence as suggested.

Line 56: “…, considered the election method for the detection 56 of toxic elements” is confusing. Authors should consider deleting.

Dear reviewer, we removed this part as suggested.

Lines 58-62: For readability, breakdown this long sentence. Perhaps the following rewrite can be of help “The aim of this work is to assess the presence of As, Cr, Cd and Pb in East Asian bullfrog (Hoplobatrachus rugulosus) legs marketed in the large-scale retail trade. This is done for three reasons: (i) to provide first data on the toxicological aspects of the edible frogs, (ii) to make an initial risk assessment on the consumption of the frogs, and to provide scientific evidence needed to raise awareness in the legislature.”

Dear reviewer we rewrote this sentence as suggested.

Line 69: Mineralisation procedure could mean just about anything. Seriously consider replacing the entire sentence with something like “The samples were digested according to protocols reported before [8,9].”

Dear reviewer we rewrote this sentence as suggested.

Line 74: Instead of “The samples mineralised were …” consider rewriting as “The digested samples were….”

Dear reviewer we rewrote this part as suggested.

Line 75: Consider changing “before the ICP-MS” to “before ICP-MS”

Dear reviewer we deleted “the” as suggested.

Line 77: Consider changing “…instrumental settings were set as following” to “…instrumental settings were as follows”

Dear reviewer we rewrote this part as suggested.

Lines 81-84: Confusing and must be rewritten. Furthermore, (i) at least in the “at least 7 points” points to uncertainty; (ii) “admitting” in line 83 (and line89) are really out of place; and (iii) what is “white calibration,” how does it apply to this study?

Dear reviewer we rewrote this part as suggested.

Line 85: Which method was validated, sample collection or ICP-MS or both?

Dear reviewer, we confirm that we validated only the ICP-MS method.

MATERIALS AND METHODS

  1. The foundation of Health Risk Assessment (HRA), as reflected in this manuscript, is confusing and shows limited understanding of risk assessment of mixed chemicals as exposed by the US EPA. There are four (4) sequential steps required to carry out a successful HAR; this knowledge is not clear in this manuscript. This reviewer recognizes this shortcoming as probably a consequence of difficulties associated with translating text from other languages to written English (see Acknowledgments line 236). But, given that the gist of this paper is about Health Risk Assessment and the THQ system, it is strongly recommended that the authors include reference to the very simple description of these two concepts, such as those provided in https://doi.org/10.1371/journal.pone.0212938 and https://doi.org/10.1016/j.toxrep.2017.03.006 for the benefit of readers encountering the concept of HRA and THQ for the first time.”

Dear Reviewer, we added these important citations as requested.

  1. Consistency is required. For example, the authors should decide whether to use:

(i)     “where” (equation 2), “Where” (line 89 and equation 3) or “Where:” (equation 4)

(ii)   “As, Cr and Cd” (Line 124) or “cadmium, chromium and arsenic” (Line 126).

(iii) “kg” (line 116), or “Kg” (line 96 and the entire manuscript). NB: The symbol of the SI unit of mass kilogram is “kg”, not Kg. This correction, in this reviewer’s opinion, must be made throughout the manuscript.

Dear reviewer, we have done all the changes requested.

  1. Remove unnecessary repetition. For example,

(i)     “BW is the average body weight (70 Kg)” is found on lines 110, 115 and 127 on the same page. Please define whose body weight is being referred to here: male, female, children, adult human, European?

(ii)   “EDI is the estimated daily intake” is found on lines 104 and 115 on the same page.

Dear reviewer, we have done all the changes requested.

  1. From equation (2), EDI is proportional to 1/BW and from equation (3), BMDL is therefore proportional to 1/BW2. Is this the message the authors are conveying?

Yes, we confirm this.

  1. The “estimated weekly intake (EWI)” in Line 104 is not a category in Table 3 as it is incorrectly stated in Line 155.

Dear Reviewer, considering that the EWI is calculated by EDI*7 we retained superfluous to put these results in the table.

Line 98: (Helsel (2005) [11]” is incomplete in the References section.

Dear reviewer, we corrected the reference as requested

Line 99: “…prior to analysis.” Are the authors refereeing to data or samples analysis?

Dear reviewer, we specify it in the main document.

LinE 106:  “… 225g of frog legs are consumed daily” Who is consuming this? It is not clear from text.

Dear reviewer, we added more details in the main document.

RESULTS AND DISCUSSION

  1. Consistency is required. For example, the authors should decide whether to use:

(i)     Small “t” or capital “T” when referring to a table in text (e.g., “table 1” & “table 2” (lines 135, 136 & 140), but Table 3” line 155.

Dear reviewer, we have done all the changes requested.

  1. Consider rewriting “The linearity test of the ICP-MS method for As,Cr, Cd and Pb gave satisfactory results” as “The linearity test of the ICP-MS method for As, Cr, Cd and Pb gave satisfactory results”

Dear reviewer, we have done all the changes requested.

  1. Table 2: (i) Replace “Site 1” and “Site 2” in table and caption with “Thailand” and “Vietnam” respectively. No need to be complicated since the expressions “Site 1” and “Site 2” are not used by the authors in the text. (ii) For clarity, consider rewriting the caption to read “The level of As, Cr, Cd and Pb (mg/kg) of samples detected in the study East Asian bullfrog. Cd was not detected in any sample. Entries n.d.= not detected.” If authors agree to adding the statement “Cd was not detected in any sample,” the entire and redundant Cd columns in Table 2 can be deleted.

Dear reviewer, we have done all the changes requested.

Lines 134-139: Consider rewriting this paragraph and moving it to the “Materials and Methods” – along with Table 1 under a separate subsection; you may call the subsection “2.3 Calibration, limits of detection and quantification.”   The authors must state briefly how LOD & LOQ are determined here.  Of note, “Linearity test” is vague; consider replacing it with “ICP-MS calibration” both in text and in Table 1 caption. Also in the caption to Table 1, define “x” and “y”. In Line 136, “Recoveries evaluation showed values between 96% and 105%”, provide what is being evaluated and reason for evaluation.

Dear reviewer, we added more information for the LOD and LOQ calculation and for recovery evaluation in the materials and methods section.

Line 147: What is “Conservely”?

Dear reviewer, it was a typing mistake, we meant

Line 149: New paragraph should begin at “The maximum Cr, As e Pb values found in the…” What is “e” between AS and Pb?

Dear reviewer, we corrected this part as suggested.

Hope these changes could be helpful for the manuscript reconsideration.

Thank you for the privilege to consider our work in this esteemed journal.

Kind regards

Round 2

Reviewer 1 Report

The addressed all my questions.